# Efficient Continuous Subgraph Matching Scheme Based on Trie Indexing for Graph Stream Processing

**Dojin Choi** [1]**, Somin Lee** [2]**, Sanghyeuk Kim** [2]**, Hyeonbyeong Lee** [2] ⬤**, Jongtae Lim** [2] ⬤**, Kyoungsoo Bok** [3]
**and Jaesoo Yoo** [2],* ⬤

[1] Department of Computer Engineering, Changwon National University, 20, Changwondaehak-ro, Uichang-gu, Changwon-si 51140, Gyeongsangnam-do, Republic of Korea; dojinchoi@changwon.ac.kr

[2] Department of Information and Communication Engineering, Chungbuk National University, 1, Chung-dae-ro, Seowon-Gu, Cheongju 28644, Chungcheongbuk-do, Republic of Korea; smlee@cbnu.ac.kr (S.L.); tiensh12369@naver.com (S.K.); lhb@cbnu.ac.kr (H.L.); jtlim@cbnu.ac.kr (J.L.)

[3] Department of Artificial Intelligence Convergence, Wonkwang University, 460, Iksan-daero, Iksan-si 54538, Jeollabuk-do, Republic of Korea; ksbok@wku.ac.kr

* Correspondence: yjs@cbnu.ac.kr; Tel.: +82-43-261-3230

**Abstract:** With the expansion of the application range of big data and artificial intelligence technologies, graph data have been increasingly used to analyze the relationships among objects. With the advancement of network technology and the spread of social network services, there has been an increasing need for a continuous query processing algorithm that can manage large-volume graph streams generated in real time. In this paper, a sliding-window-based continuous subgraph matching algorithm that can efficiently control graph streams is proposed. The proposed scheme uses a query processing technique based on trie indexing. It establishes an index based on a materialized view of similar queries and conducts continuous query processing based on the materialized view to perform continuous query processing efficiently. It also provides wildcard operations on vertices and edges to consider various query types. Moreover, in this study, a two-level cache technique that can manage frequently used subgraphs and subgraphs that may be used in the future is developed, to handle intermediate query results in the form of a materialized view. Cache replacement techniques based on statistical data are also presented to improve the performance of the developed cache technique. The excellent performance of the proposed algorithm is verified by a conducting independent performance evaluation and comparative performance evaluation.

**Keywords:** continuous query processing; stream processing; graph stream; subgraph matching; materialized view

## 1. Introduction

Graph data are used to perform the modeling of interactions among objects in various fields, such as social networks, communication networks, transport networks, and knowledge graphs [1–6]. Graph data can effectively represent multiple relationships among objects based on vertices and edges and show the characteristics of vertices and edges based on attributes and labels. With the expansion of the range of fields applying big data and artificial intelligence, graph data have been increasingly used to obtain insights into complexly connected data. Graphs are also used to detect or analyze the changes in relationships among objects and derive effective results for fraud detection, anomaly detection, topic detection, and service recommendation.

Graph data have increased in volume and are modified in real time owing to the development of network technology and the spread of services associated with the Internet of Things and social networking. These phenomena have recently contributed to the creation of graph streams in which vertices and edges that compose graph data change in real time. For example, social networking services, such as Twitter and Facebook, show

a constant change in personal relationships among users. In social networking service environments, these users also create and share various types of documents. Consequently, the elements of graphs that are required for social network analysis are continuously altered.

Researchers have used diverse graph analysis algorithms to extract significant information from graph streams that change in real time. Representative graph analysis algorithms include shortest path, PageRank, pattern analysis, frequent pattern detection, and subgraph matching algorithms [7–12]. A subgraph matching algorithm identifies a similar subgraph to a query graph defined by the user in graph data [13]. This algorithm is used to find a subgraph that has a similar pattern to that of a query, among data graphs in numerous applications that deal with graphs, such as those of protein interaction networks and social networks [14–19]. It is classified into two types: static subgraph matching algorithms [14–16], which perform subgraph matching based on large-volume graphs defined in advance, and continuous subgraph matching algorithms [17–19], which perform subgraph matching based on graphs that change constantly over time. A static subgraph matching algorithm generates a similar subgraph to a query as an outcome under the condition where the entire graph data are stored in advance. A continuous subgraph matching algorithm compares patterns of dynamically changing graph streams similar to those of queries based on a specific time interval in real time and generates adjusted results.

As graph data have tended to be inputted in the form of streams in most recent applications, researchers have actively examined continuous query processing for real-time graph stream processing [17–20]. The method in [17] based on Pregel enables users to perform distributed processing easily when they define a function to be executed by a vertex. However, this method has a problem in that all the graphs should be contained in the memory [18]. The method in [19] does not store the entire graph data but stores only information on the state of transfer of vertices and the occurrence of edges based on graph streams. However, as this method stores information on the state of transfer of vertices according to queries, it cannot handle similar queries at once. Moreover, it does not provide a parameter query as a query type, although it uses a data structure considering query types. It does not present an additional caching technique either. The method in [20] efficiently performs multiple subgraph matching tasks by using trie indexing. This method stores the intermediate query results in the form of a materialized view, thereby facilitating swift query processing. However, it does not present a specific cache application measure. Existing studies assumed that all the intermediate results of indexing or query processing are resident in the memory. However, as a massive amount of graph data are generated in practical applications, the entire graph data are unlikely to be stored in a limited memory environment. To solve this problem, a method that can efficiently perform continuous subgraph query processing based on graph streams needs to be developed by considering a limited memory environment.

In this paper, a sliding-window-based continuous subgraph matching scheme that can efficiently perform graph stream processing is proposed. The proposed scheme uses trie indexing, which can handle similar queries at once. The trie indexing technique integrates indexes for similar paths into a single index based on a principle that similar queries in a graph can be divided according to similar paths. Hence, this scheme is adopted in this study in continuous query processing. The proposed continuous subgraph matching algorithm also provides wildcard operations based on both vertices and edges to consider various query types. Accordingly, a statistics-based two-level caching technique that can efficiently manage intermediate query results renewed by graph streams is presented herein. This technique facilitates efficient continuous subgraph query processing by managing frequently used data at the top level and placing intermediate query results, which can be used for the following query processing, at the lower level. Furthermore, various cache replacement strategies are introduced to increase the efficiency of the existing trie indexing structure. This study has the following contributions:

1.  Trie-index-based similar query index management: Stream processing requires a quick response to queries because graph data is constantly entered. The proposed scheme improves the processing time of continuous queries by judging structurally similar graph queries as similar queries and managing similar queries with the trie index.

2.  Support for various query types: The proposed scheme supports various types of graph queries. As in previous studies, it additionally supports wildcard operations as well as labels of vertices and edges. In the proposed study, wildcard means a query request that can be any label of the vertex or edge. Consideration of these wildcard operations was additionally made to manage the index of similar queries in the above-mentioned contribution.

3.  Statistics-based two-level caching technique: The proposed study presents a statistics-based two-level caching technique to perform real-time stream processing. The proposed caching technique improves the query processing speed by caching data frequently used for query processing and data to be used for query processing in advance in-memory. The existing two-level caching technique proposed by this research team was modified to a form that can be applied to the trie index. In the real world, many queries can be registered and added continuously. If so, all indexes cannot be loaded into the memory, so the indexes must remain on the disk, and only the indexes currently required to process the query must be able to reside in the two-level cache. We verify the validity of the proposed statistics-based two-level caching technique through performance evaluations.

The remainder of this paper is organized as follows. Section 2 reviews related previous studies and describes their limitations. Section 3 describes the details of the continuous subgraph matching algorithm proposed herein. Section 4 verifies the excellent performance of the proposed algorithm based on performance evaluation results. Finally, Section 5 presents the conclusions and future research directions.

## 2. Related Work

A materialized view is a database object that contains the results of a query and the query processing results. When a person conducts an operation, such as the aggregation or search of large-volume data, he or she may need to combine several tables to complete this operation. This type of complex operation may require a considerable amount of time for query processing. At this moment, a materialized view that stores query results in physical space can be implemented to reduce query processing loads. A general view maintains only query processing methods and does not generate practical query results. On the contrary, a materialized view contains practical query results in a table. That is, a general view is a logical table, whereas a materialized view is a table in physical space. A previous study developed a continuous subgraph matching algorithm based on a directed acyclic graph (DAG) to perform continuous subgraph matching in a real-time graph stream [17]. This algorithm converts a query graph to a DAG and detects a sink vertex that includes only ingoing edges in the converted DAG. Based on the detected sink vertex, it divides a query into subDAGs. Then, it detects a source vertex (root vertex) that includes only outgoing edges in each subDAG. Each source vertex sends a message to vertices that contain the same label values as those of a query, among the vertices connected to it. When a sink vertex receives a message from all the outgoing edges connected to it, the DAG-based continuous subgraph matching algorithm generates subgraph matching results. Subsequently, this algorithm performs final subgraph matching by combining the intermediate results of subDAGs. This algorithm uses Pregel, a graph distribution processing model, to transfer messages [21]. Pregel defines a function to be executed by a vertex in a graph, calls a function where all the vertices are user-defined in each superstep, and performs the next synchronization process in the order. Since a DAG-based continuous subgraph matching scheme uses Pregel, distributed processing is easy if a user defines only the functions to be performed by the vertex. However, these distributed systems must keep all data graphs

in their memory when computing graphs. In addition, it is difficult to use in a limited memory environment because the messages necessary for communication between each node of the distributed server must be kept in the memory.

Early continuous subgraph matching algorithms generated a redundant indexing cost when similar queries were registered. To solve this problem, a continuous subgraph matching algorithm based on a transition state was presented [19]. This algorithm defines a transition state of vertices in a record and records the state of vertices according to the occurrence of edges in a query graph. When all the vertices satisfy a certain transition state, this algorithm determines this state to be the final subgraph matching state. It uses a renewal-type graph that contains the vertices of a candidate query for the vertices of a data graph to store information on partial consistency with a query graph. This type of data-centric representation (DCG) is applied to maintain intermediate query results. When a graph stream is updated, information on consistency with a query graph is stored by an edge transition model by applying DCG. Each edge has a state among explicit, implicit, and null states. As for graph stream data, the transition state of vertices is stored according to query types. The state of an edge is stored as a null state when data that match with edge types do not exist. The state of an edge is stored as an implicit state when data match with the edge type of a query but do not match with the edge type of lower nodes. The state of an edge is stored as an explicit state when data match with both the edge types of a query and lower nodes. When all the edges appear and are in an explicit state, the continuous subgraph matching algorithm based on a transition state generates this result as the final subgraph matching result. Transition state-based continuous subgraph matching scheme has low index costs because they manage DCGs while storing only vertex transition state and edge appearance information for streams [18]. However, there is a limit to keeping all DCGs in the memory for all data graphs. In addition, since the transition state is stored by query, similar queries cannot be processed at once. Therefore, in an environment with many similar queries, the query processing speed may be significantly reduced.

Another study presented a continuous subgraph matching algorithm containing intermediate query results in the form of a materialized view to encourage various queries in the continuous subgraph matching process [20]. This algorithm consists of a query indexing phase and a query answering phase. In the query indexing phase, this algorithm obtains a covering path through query graph decomposition. A covering path is a set of the shortest paths including all the vertices and edges of a query graph. Then, this algorithm builds a set of shared queries based on trie query indexing. This algorithm can efficiently perform multiple query processing tasks, as it responds to queries based on a shared common pattern at once. It also conducts subgraph matching based on trie traversal and join operations. In the query answering phase, this algorithm seeks the list of tries affected by the update of a graph stream. It performs join operations based on trie traversal and updates a materialized view. It repeats this process until it reaches child nodes. Then, it terminates the query search process at a subtree not affected by the update of a graph stream. However, Ref. [20] proposed a technique that added a cache, but it did not provide a detailed replacement cache strategy. For more efficient subgraph matching, a cache replacement technique considering an index structure is required. It also requires support for various wildcard operations.

Ref. [22] addressed the problem of answering graph pattern queries using materialized views based on large data graphs under homomorphisms. It considers a broad class of pattern queries that involve both node reachability and direct relationships. It treats materialized views as summary graphs to represent the homomorphic matches of the views compactly. However, it does not cover the stream processing environment.

Ref. [23] presents approximating simulation and subgraph queries using views. They have proposed a notion of upper and lower approximation of pattern queries with respect to a set of views. It develops efficient exact and approximation algorithms with provable bounds for computing the closest upper and lower approximations, complete or not. They

use views for subgraph query answering efficiently. Similarly, this study does not deal with graph stream processing. This study focuses on approximate query processing.

## 3. Proposed Continuous Subgraph Matching Scheme

### 3.1. Overall Structure

In this paper, a continuous subgraph matching scheme that can solve the problems of existing continuous subgraph matching algorithms is proposed. In the proposed method, a query refers to a request for subgraph matching. When a graph stream is input, the proposed method performs a filter-then-verify (FTV) process based on trie indexing to respond to a query on continuous subgraph matching. It is more efficient to handle the same or similar queries at once than to handle each registered query in the process of managing queries on continuous subgraph matching based on a graph stream updated in real time. Hence, the proposed algorithm indexes a common pattern of queries by considering the redundancy of queries. It performs continuous subgraph matching by comparing a graph stream updated in real time with indexed queries. The intermediate query results should be contained in the memory and used owing to the characteristics of continuous query processing. To reuse intermediate query results efficiently, the proposed method generates intermediate query results in the form of a materialized view and manages these results based on a cache.

Figure 1 shows the overall system structure of the proposed method, which consists of an Index Manager, Cache Manager, and Query Processing Manager. The Query Processing Manager comprises a stream processing module that performs graph stream processing and a subgraph matching module that performs subgraph matching in response to queries. The Index Manager conducts the process of indexing a query graph, called query indexing, in advance. Trie query indexing is applied to respond to multiple subgraph matching queries efficiently. The Query Processing Manager conducts continuous subgraph matching based on a graph stream updated in real time. The stream processing module extracts a set of candidates for subgraph matching based on a graph stream. This module uses existing intermediate results loaded in a cache to execute query processing efficiently. The subgraph matching module performs join operations by using a set of candidates for subgraph matching and the intermediate results loaded in a cache. Through join operations, a material view that represents intermediate results is generated. Based on the materialized view, the proposed method performs final subgraph matching. The Cache Manager conducts query processing based on a two-level caching technique and constantly records statistical data based on the Statistics Manager. When the space of a cache is insufficient, the statistics-based cache replacement strategies proposed herein are adopted to replace data in a cache.

In this study, a graph is defined as G = (V, E, L(V), L(E)) where a direction and label are applied. V is a set of vertices, E is a set of edges, L(V) is a set of labels of vertices, and L€ is a set of labels of edges. Each vertex (v∈V) has a unique number (v_id) and unique label (v_label). Each edge (e∈E) is a pair of vertices (e = (s, t)). Here, s is a root vertex (source vertex), and t is an end vertex (target vertex). A query graph is defined as $Q_i$ = ($VQ_i$, $EQ_i$, L(V), L(E)). A graph stream is input based on the unit of a sliding window. It is assumed that a graph stream is input in the form of an edge stream ($v_1$_id, $v_2$_id, $v_1$_label, $v_2$_label, and e_label, *t*). The unique numbers and labels of each vertex and the labels of each edge are input. Here, t refers to the time of generation of a graph stream.

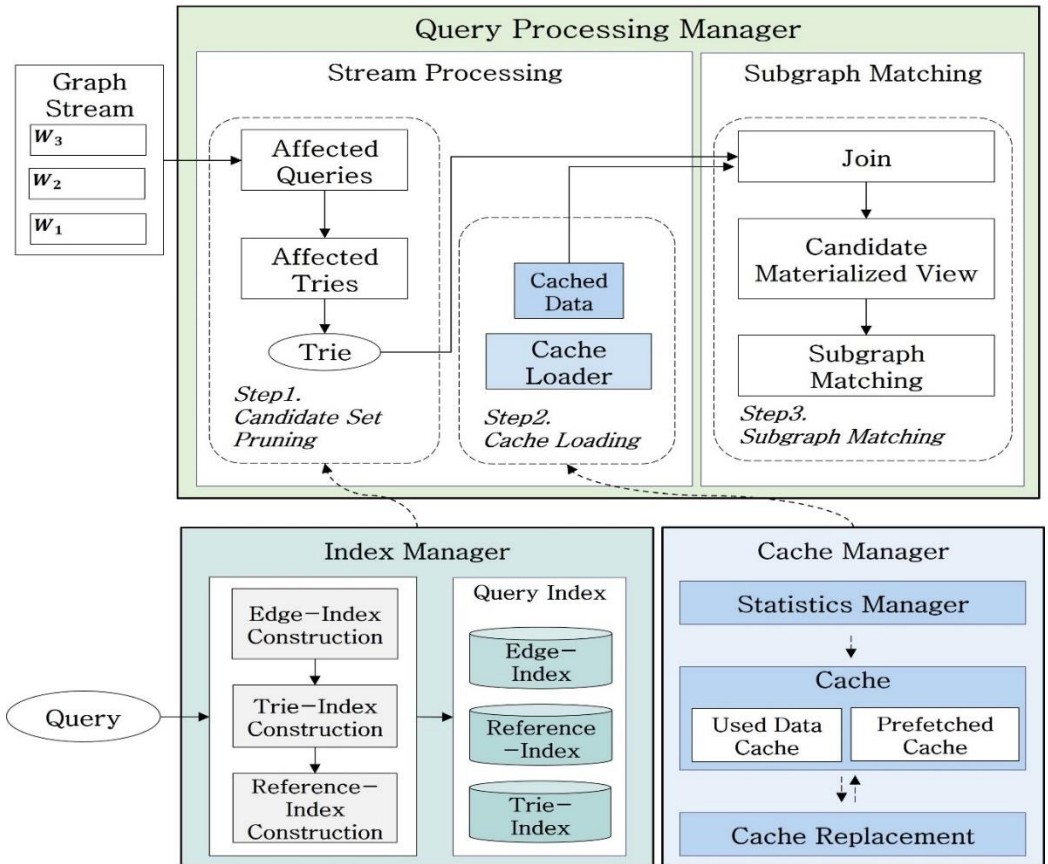

**Figure 1.** Overall system structure.

### 3.2. Index Manager

A uniform response to a common pattern of queries is an efficient method for multiple subgraph matching processing tasks. The Index Manager is used to analyze patterns of duplicated queries and index a duplicated query area. When common patterns are indexed at once, the cost of indexing new similar queries can be reduced. As the Index Manager calculates query processing results in a duplicated area, it also reduces the query processing cost. The query decomposition and indexing technique proposed by [19] is used in this study. The Index Manager divides a query into max covering paths, which are a set of the shortest paths including all the vertices and edges of a query graph. A query $Q_i$ is divided into k paths ($MCP(Q_i) = \{P_1, P_2, \ldots, P_k\}$). The edge indexing process records information on a query ($Q_i$) including the corresponding edges of each path to determine a query affected by an input graph stream during query processing. The trie indexing process stores the edges of each path in trie nodes. Each node maintains its unique node number and information on parent nodes and child nodes. This process conducts join operations based on the parent and child nodes of the corresponding node during query processing to generate query results. The reference indexing process records reference information on the last terminal node in a path stored in the trie indexing process according to queries. The Index Manager detects a query affected by the edge indexing process and identifies the location of the node that contains the corresponding edges based on reference information. Based on the method described above, the Index Manager records queries with the same path in the same trie node and uses shared information on similar queries to respond to multiple queries efficiently.

Figure 2 shows the process of indexing a query graph $Q_1$. $Q_1$ is divided into three max covering paths ($MCP(Q_1) = \{P_1, P_2, P_3\}$). For example, $P_1$ indicates the following three edges: $e_1 = <v_1, var>$, $e_3 = <var, v_2>$, and $var = <v_2, v_4>$. Here, var refers to a wildcard operation. This operation indicates that information on vertices and edges has not

been specified and that vertices or edges can be randomly connected. The edge, trie, and reference indexing processes are applied to the max covering paths. In the edge indexing process, information on each edge of MCP($Q_1$) and the current query ($Q_1$) included in the information on edges ($e_1 = <v_1, var>$, $e_3 = <var, v_2>$, and $var = <v_2, v_4>$) are stored in $P_1$. The trie indexing process stores edges included in each path in trie nodes. The edges ($e_1 = <v_1, var>$, $e_3 = <var, v_2>$, and $e_2 = <v_2, v_4>$) included in $P_1$ are stored in trie nodes $n_1$, $n_2$, and $n_3$. The reference indexing process stores a reference on the last location of a path and stores the reference information on a node ($n_3$) storing the last edge of $P_1$. Furthermore, $e_1 = <v_1, var>$, an edge found in each path of $Q_1$ in common, is stored in a node to facilitate path exchange.

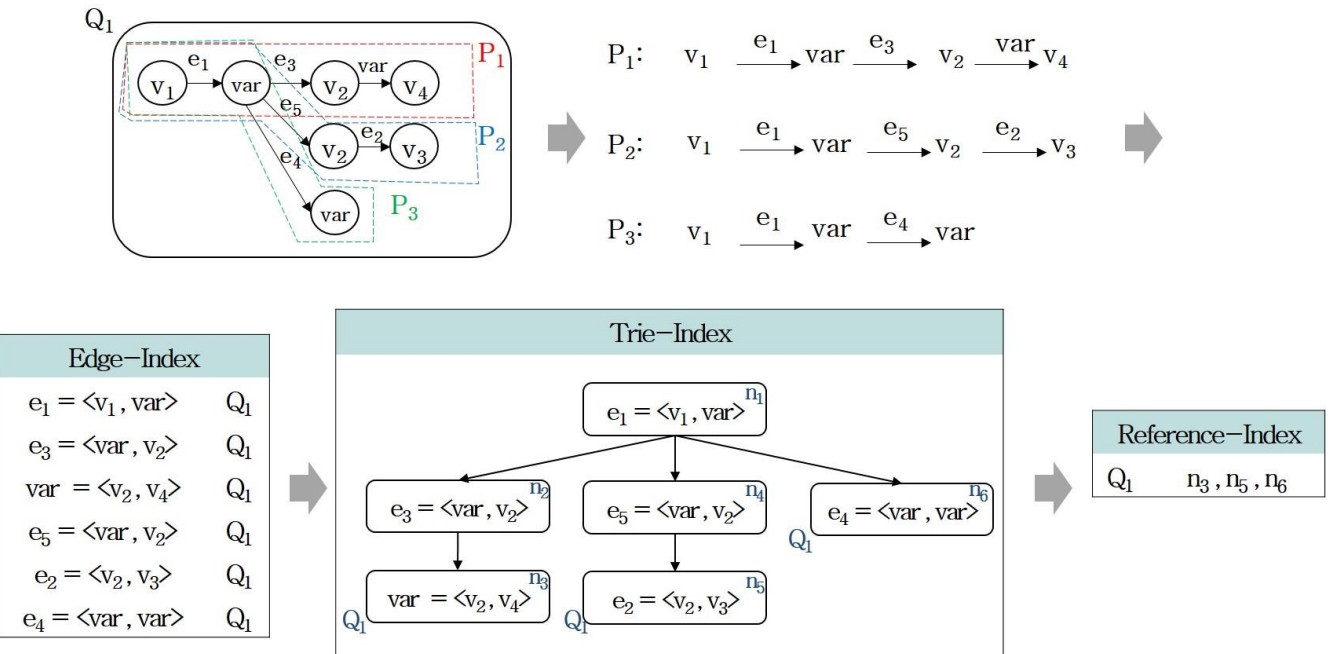

**Figure 2.** Process of indexing a query graph $Q_1$.

Figure 3 shows a query graph $Q_2$ input under the condition where a query graph $Q_1$ is indexed. Like $Q_1$, $Q_2$ can also be divided into max covering paths. Consequently, $Q_2$ is divided into two max covering paths (MCP($Q_2$) = {$P_1$, $P_2$}). When the edge indexing process indexes each max covering path, information on $Q_2$ is added to existing information on edges $e_1 = <v_1, var>$ and $e_4 = <var, var>$. Accordingly, a new edge ($e_2 = <v_1, var>$) is added. The trie indexing process stores the edges of each path in trie nodes. When the path of $Q_2$ is equivalent to the existing path of $Q_1$, the corresponding trie node is shared. For example, $P_1$ ($e_1 = <v_1, var>$, $e_4 = <var, var>$) is equivalent to $P_3$, the path of $Q_1$. Consequently, the same trie node is shared. A newly created path is stored in the root node. Furthermore, $e_2 = <v_1, var>$, the edge of $P_2$, is stored in a new node $n_7$. In the reference indexing process, reference information {$n_6$, $n_7$} on the last terminal nodes in each path is added.

As the trie indexing technique shares the path observed in each query in common it can lead to the efficient indexing of similar queries. Figure 4 shows the indexing structure based on queries $Q_1$ to $Q_5$, including query graphs $Q_1$ and $Q_2$. The query indexing processes consist of trie indexing, edge indexing, and reference indexing.

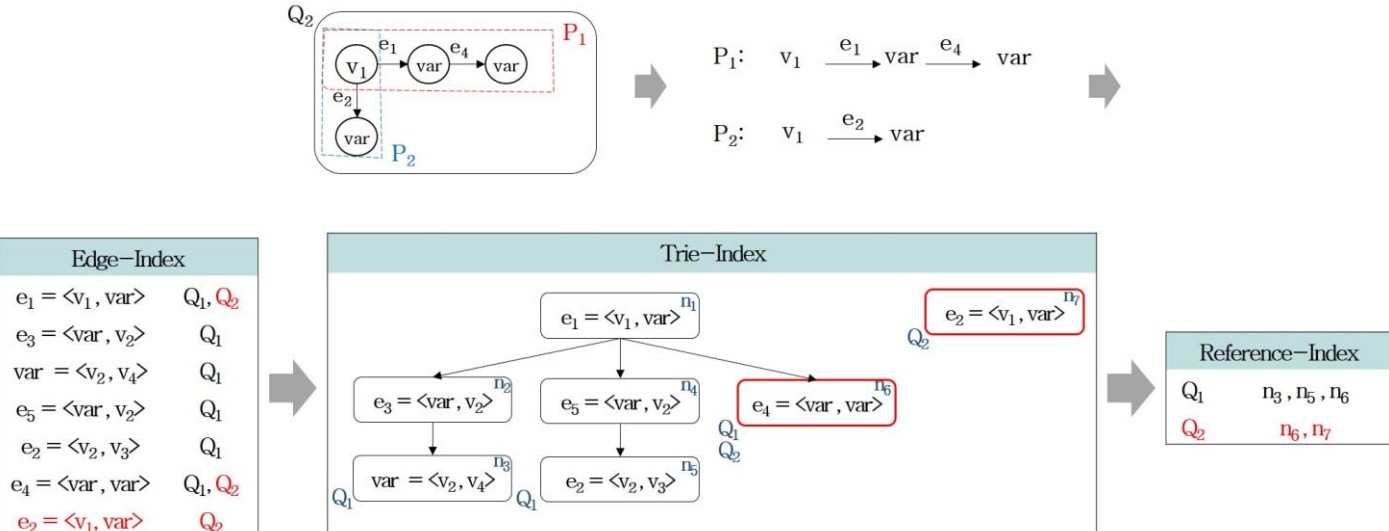

**Figure 3.** Process of indexing a query graph $Q_2$.

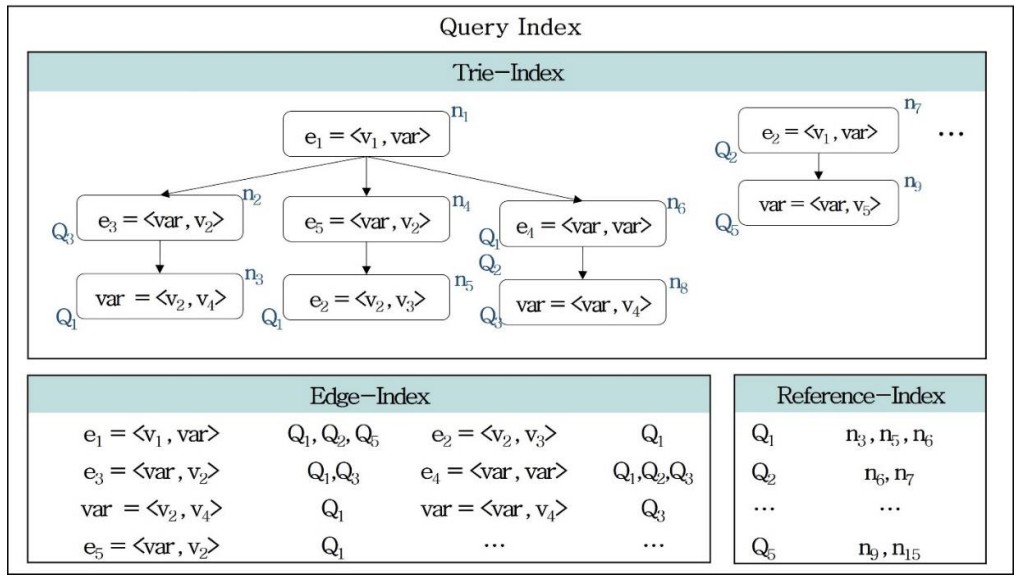

**Figure 4.** Indexing of query graphs $Q_1$ to $Q_5$.

### 3.3. Cache Manager

In this study, a two-level caching technique [24] is developed to perform continuous subgraph matching efficiently in a limited memory environment. The developed algorithm is applied to facilitate efficient subgraph query processing. Existing intermediate results stored in a cache are used to perform query processing efficiently. A used cache manages intermediate query results that are frequently used in query processing, and a prefetched cache manages intermediate query results that are likely to be used for the next query processing task. To apply [24] to this study, the trie indexing results and intermediate results were adjusted to be indexed. Accordingly, a prefetched cache was adjusted to manage the trie indexing results that are likely to be accessed. As for intermediate query results, information on the corresponding edges and information on results are separated for management. The information on edges includes the unique numbers of the corresponding edges recorded, whereas the information on results includes the unique numbers of join operation results. As the root trie node is regarded as results, both the information on edges and the information on results are updated to contain information on this node. When a target node is not the root trie node, the results on the current trie node are updated based

on the information on the edges of the target node, the results on the parent trie node, and the results of the join operation. Moreover, the results on the child trie node are repeatedly updated based on the results on the current trie node, the information on the edges of the child trie node, and the results of the join operation.

Figure 5 shows the operation of the Cache Manager. Figure 5a is an example of the first window. When a graph stream $<0, 1, v_1, v_2, e_1, 1>$ is input, this stream is compared with the trie indexing results to determine a candidate set $\{n_1\}$ for subgraph matching. As the trie node $n_1$ is the root node, a prefetched cache loads intermediate results on the child trie nodes $\{n_2, n_4, n_6\}$ of $n_1$ for query processing. These results are not loaded in the current cache, which does not contain any data. In the subgraph matching process, the Cache Manager updates information on a used cache and relevant statistical information. The information on edges for a used cache includes the unique numbers of trie nodes and the unique numbers of the vertices of edges used for query processing. Consequently, $(n_1, (0, 1))$ is recorded. As $n_1$ is the root node, it is regarded as query results. Consequently, the result on n1 (0, 1) is applied as intermediate query results in the form of a materialized view. Statistical information comprises the unique number of a trie node, load time, last hit time, the number of hits, the number of joins, and the size of the intermediate result. As for statistical information on n1, the Statistics Manager records a load time as the current window time 1 and the size of intermediate results as 2. Figure 5b is an example of the second window. Like the processes described above, a candidate set $\{n_6\}$ is determined based on an input stream. To perform the processing of a trie node $(n_6)$ instead of the root node, intermediate results on the parent trie node $(n_1)$ and the child trie node $(n_8)$ in the form of a materialized view are loaded in a prefetched cache. A prefetched cache is used to load the data to be used for the next query processing in advance. Hence, when a used cache exists, it is instantly used. In (a), $n_1$ is loaded in a cache while it is used for query processing. Hence, it is resident in a used cache. Then, subgraph matching based on the loaded data is conducted. The result on $n_6$ (0, 1, 2) is updated as the intermediate result in the form of a materialized view based on the result on the parent trie node $(n_1)$ loaded in a cache and join operation results. As for statistical information on n6, the Statistics Manager records a load time as the current window time 1 and the size of intermediate results as 3. It also updates statistical information on n1 by applying the last hit time as 1, the number of hits as 1, and the number of joins as 1. When the size of the data to be loaded exceeds the size of the cache, cache replacement based on statistical information and a cache replacement strategy is performed.

When graph data are input, the Query Processing Manager undergoes the FTV process based on trie indexing to implement query processing. To perform query processing efficiently, it uses intermediate results loaded in a cache. The Cache Manager uses the Statistics Manager to record the load time of a cache, the number of hits, the number of joins, and the size of intermediate results constantly. When the space of a cache is insufficient, the statistics-based cache replacement strategy is adopted to replace the data in a cache.

When a cache of limited capacity becomes full of necessary subgraphs for query processing, the subgraphs stored in a cache should be replaced. At this time, the Cache Manager manages statistical data according to subgraph matching query processing and conducts a cache replacement based on the updated statistical information. Table 1 shows an example of a statistical table. ID is the unique number of a node, load time (LT) is the time of data loading, last hit (LH) is the last time a cache was hit, the number of hits (H) is the number of successful hits, the number of joins (J) is the number of successful join operations, the result size (S) is the scale of query results, and the query size (QS) is the scale of a query.



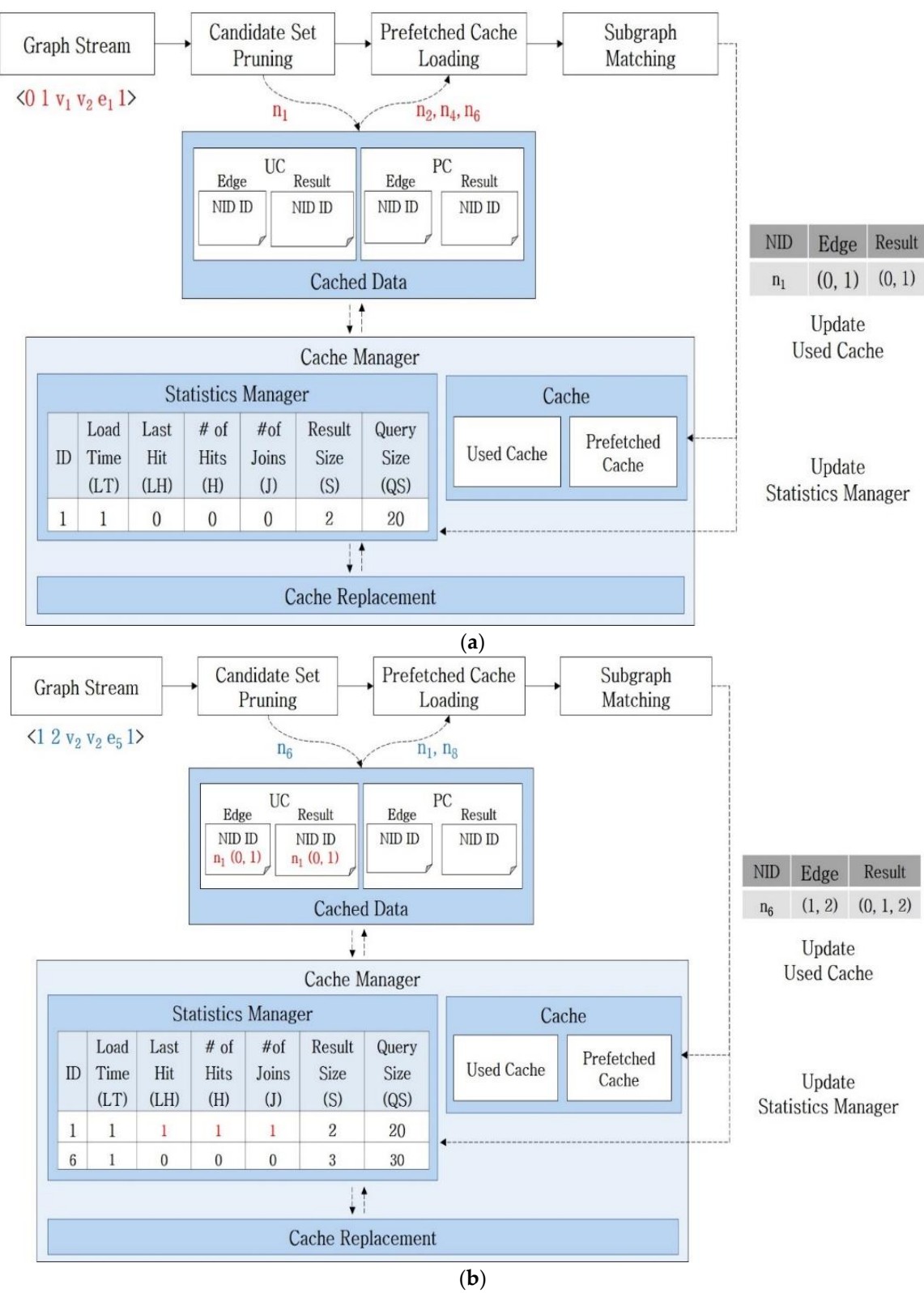

**Figure 5.** Examples of cache data management. (**a**) Graph stream $SW_1$; (**b**) Graph stream $SW_2$.

**Table 1.** Table of cache statistics.

| ID | Load Time (LT) | Last Hit (LH) | # of Hits (H) | # of Joins (J) | Result Size (S) | Query Size (QS) |
|---|---|---|---|---|---|---|
| 1 | 0 | 50 | 100 | 90 | 100 | 20 |
| 2 | 5 | 67 | 72 | 10 | 80 | 10 |
| 3 | 10 | 60 | 63 | 23 | 30 | 70 |
| 4 | 20 | 93 | 50 | 25 | 50 | 100 |
| 5 | 99 | 100 | 1 | 1 | 10 | 10 |

S is the sum of the size of all the intermediate results, and QS is the result of multiplying the number of vertices, edges, and unique labels included in a query. For example, the LT, LH, and H of the node 1 were 0, 50, and 100, respectively. The J, S, and QS of this node were 90, 100, and 20, respectively.

For example, when the least recently used (LRU) policy is applied in a situation where two caches should be replaced, the results on node 1 and node 3 will be replaced. However, this decision leads to a situation where node 1 is replaced owing to a high hit rate and where node 5, whose H is only 1, is resident in a cache. Consequently, the efficiency of a cache is likely to decrease. To increase the efficiency of a cache, the following four cache replacement strategies are proposed herein based on statistical information: the popularity (POP); popularity and usage (POU); popularity, usage, and size (PUS); and popularity, usage, size, and query size (PUSQ) strategies. In this study, the table of statistical information indicated in Table 1 is used to select the targets to be replaced by the proposed strategies. The POP strategy replaces the low result of a node derived by dividing H by the time of residence of data in a cache as shown in Equation (1). Here, i is a trie index node stored in a cache, $H_i$ is the number of hits, and $A_i$ is the time of residence of data in a cache (current time–LT). The POU strategy replaces the low result of a node derived by multiplying the value obtained by the POP strategy by a ratio of joins ($J_i/H_i$), as shown in Equation (2). $J_i$ is the number of join operations, and $H_i$ is the number of hits. A large number of join operations compared with the hit rate indicates that join operations are likely to be performed for the next graph stream. In this regard, the POU value is calculated by considering this condition. The PUS strategy replaces the low result of a node derived by multiplying the value obtained by the POU strategy by a relative ratio of S ($S/S_{max}$). Here, $S_{max}$ is the greatest result size and $S_i$ is the result size. In Table 1, the S (100) of Query 1 is $S_{max}$. When a cache includes a considerable amount of intermediate query results, this algorithm is adopted to facilitate the maximum residence of the intermediate query results in the cache. The PUSQ strategy replaces the low result of a node derived by multiplying the value obtained by the PUS strategy by a relative ratio of QS ($QS/QS_{max}$), as shown in Equation (4). Here, $QS_{max}$ is the highest QS, and $QS_i$ is QS. In Table 1, the QS (100) of Query 4 is $QS_{max}$. As indicated in the case of the PUS strategy, the PUSQ strategy facilitates the residence of data in a cache that shows high QS to increase the filtering effect.

$$POP_i = H_i/A_i, \tag{1}$$

$$POU_i = POP_i \times J_i/H_i, \tag{2}$$

$$PUS_i = POU_i \times S_i/S_{max}, \tag{3}$$

$$PUSQ_i = PUS_i \times QS_i/QS_{max} \tag{4}$$

Figure 6 shows the processes of the four cache replacement strategies developed in this study. The Statistics Manager applies each strategy based on the table of statistical information indicated in Table 1. If the POP strategy is applied for calculation under the assumption that the current time is 100, the scores for Nodes 1, 2, 3, 4, and 5 are calculated

to be 1, 0.75, 0.7, 0.625, and 1, respectively. Consequently, the POP strategy selects Nodes 3 and 4, which obtained the lowest scores, as the target nodes for replacement. At this time, Node 5's residence in the cache can decrease the efficiency of the cache. When the POU strategy is applied for calculation, Nodes 2 and 3 are selected as the target nodes for replacement. When the PUS algorithm is applied for calculation, Nodes 3 and 4 are selected as the target nodes for replacement. When the PUSQ strategy is applied for calculation, Nodes 2 and 5 are replaced. Through the aforementioned processes, Node 2, which shows a comparatively low efficiency for join operations, and Node 5, which shows a small size of intermediate results, are replaced in the cache. The Cache Manager performs subgraph matching on a graph stream and updates the cache and the Statistics Manager. Then, it applies each cache replacement strategy to the updated table of statistical information to calculate statistical values.

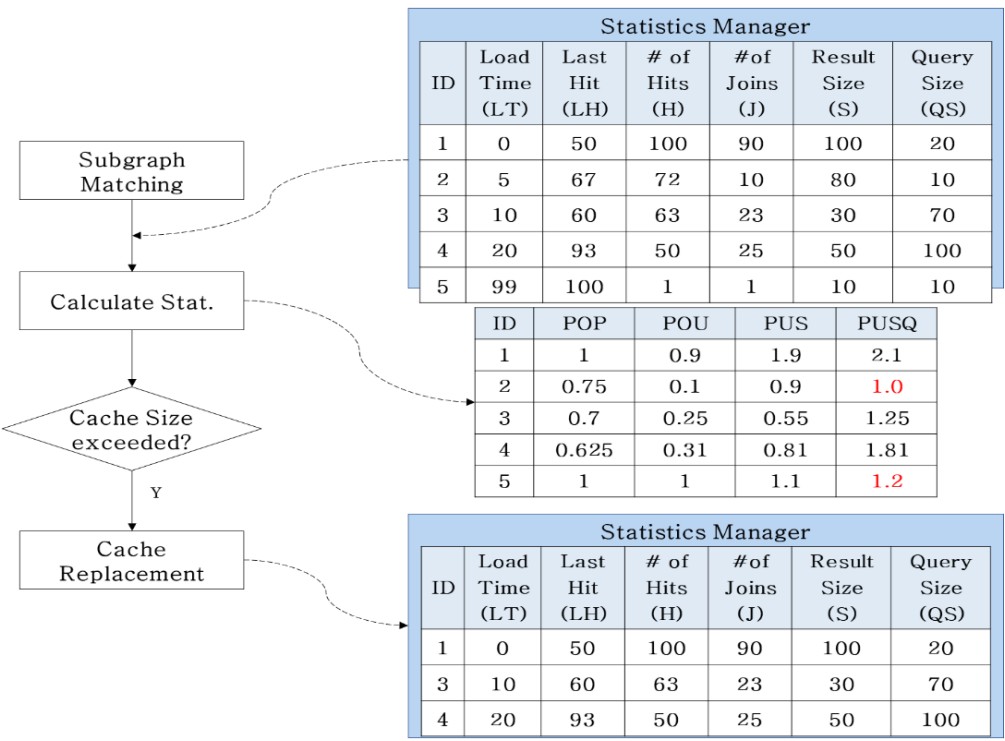

**Figure 6.** Processing processes of cache replacement strategies.

*3.4. Query Processing Manager*

The Query Processing Manager executes different processes according to the input types of stream data. If graph data is input, it analyzes if the graph data can be included in the query results managed by the current system. If the graph data can be included in the query results, it additionally generates query results. If a query graph is input, a query index is created according to the procedures described above. Figure 7 shows the procedures for query processing. When a query is requested, the Index Manager divides the query into max covering paths and creates a trie index based on each path of the query. The Query Processing Manager establishes a trie index by considering wildcard (var) operations on the query graph. In a trie, var vertices and edges are regarded and treated as nodes like vertices and indexes, including labels. When a graph stream is input after the establishment of a trie index, the stream processing module extracts a set (information on trie nodes) of candidates for subgraph matching based on the input stream. At this time, the Query Processing Manager inspects if the data of a set of candidates exist in a prefetched cache or a used cache prior to performing subgraph matching. If such data are not loaded in caches, the Cache Manager loads intermediate results on the parent and child nodes in a prefetched cache. The subgraph matching module performs subgraph matching based on

join operations by using intermediate query results loaded in a prefetched cache or a used cache. The Query Processing Manager responds to subgraph matching queries, whereas the Statistics Manager of the Cache Manager constantly records statistical information, such as the hit rate, the ratio of join operations, and the size of intermediate results. As for cache data, a TTL(Time-To-Live) value is determined based on statistics. After performing subgraph matching, the Query Processing Manager analyzes if the data exceeded the size of the cache. If the size of the cache is still sufficient, the Query Processing Manager outputs subgraph matching results. If the size of the cache is not sufficient, the Query Processing Manager replaces the data in the cache based on statistics and outputs subgraph matching results.

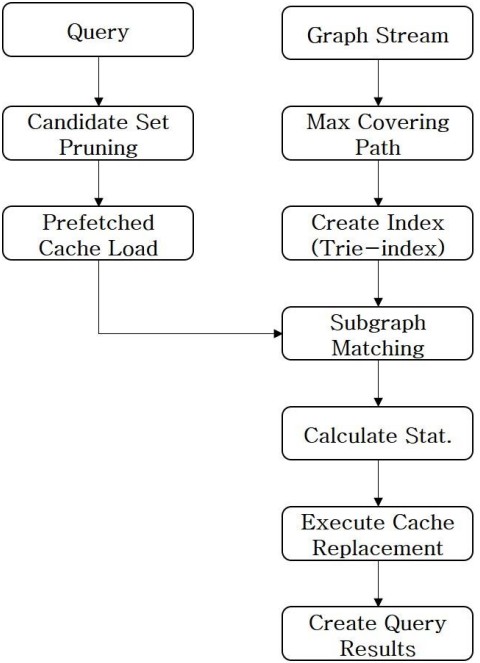

**Figure 7.** Procedures of query processing.

Figure 8 shows the subgraph matching processes. The solid line indicates query processing, and the dotted line indicates cache data management. The first edge management process on a graph stream is marked in red, and the second edge management process is marked in blue. When the edges of graph data <0, 1, $v_1$, $v_2$, $e_1$> are input, the edge indexing process is implemented to identify the list {$Q_1$, $Q_2$, $Q_5$} of queries affected by edges. The reference indexing process determines the information on trie nodes, which indicates each query stored in a trie. For example, $Q_1$ is {$n_3$, $n_5$, $n_6$}. The trie indexing process detects the location of a trie node including the indexed information on current edges <$v_1$, $v_2$, $e_1$> by performing trie node traversal according to queries. As for a specific example, the trie indexing process determines a trie {$n_1$} practically affected by edges based on the result of trie node traversal on $Q_1$. A join operation on the parent and child trie nodes of the corresponding node is conducted to generate intermediate node results in the form of a materialized view. The materialized view includes the unique numbers of nodes and vertices recorded. As n1 is a node of a current edge that serves as the root node, this node is regarded as query results. Thus, ($n_1$, (0, 1)) is recorded in the result on the materialized view of $n_1$. If the current cache does not include any data, the result of a join operation on a trie node n1 and its child trie nodes {$n_2$, $n_4$, $n_6$} is not generated. The Query Processing Manager conducts subgraph matching by repeatedly applying the aforementioned processes to all the trie nodes included in a set of candidates. When edges <1, 2, $v_2$, $v_2$, $e_5$> are input, the Query Processing Manager obtains {$n_6$} from a trie based on the same processes indicated above. A trie node $n_6$ performs a join operation with its parent trie node n1 and its child trie node $n_8$ based on intermediate results in the form of a

materialized view. As the result (0, 1) of the parent trie node n1 is loaded in the current cache, a join operation with the parent trie node generates a join result (0, 1, 2) of the trie node $n_6$. As a join operation with the child trie node $n_8$ does not generate any result, the result of $n_8$ is not created. In addition, the reference indexing process generates results of trie nodes according to queries (the results of $n_3$, $n_5$, and $n_6$ in the case of $Q_1$). When the results are generated, the Query Processing Manager conducts a join operation between trie nodes to generate the final subgraph matching results.

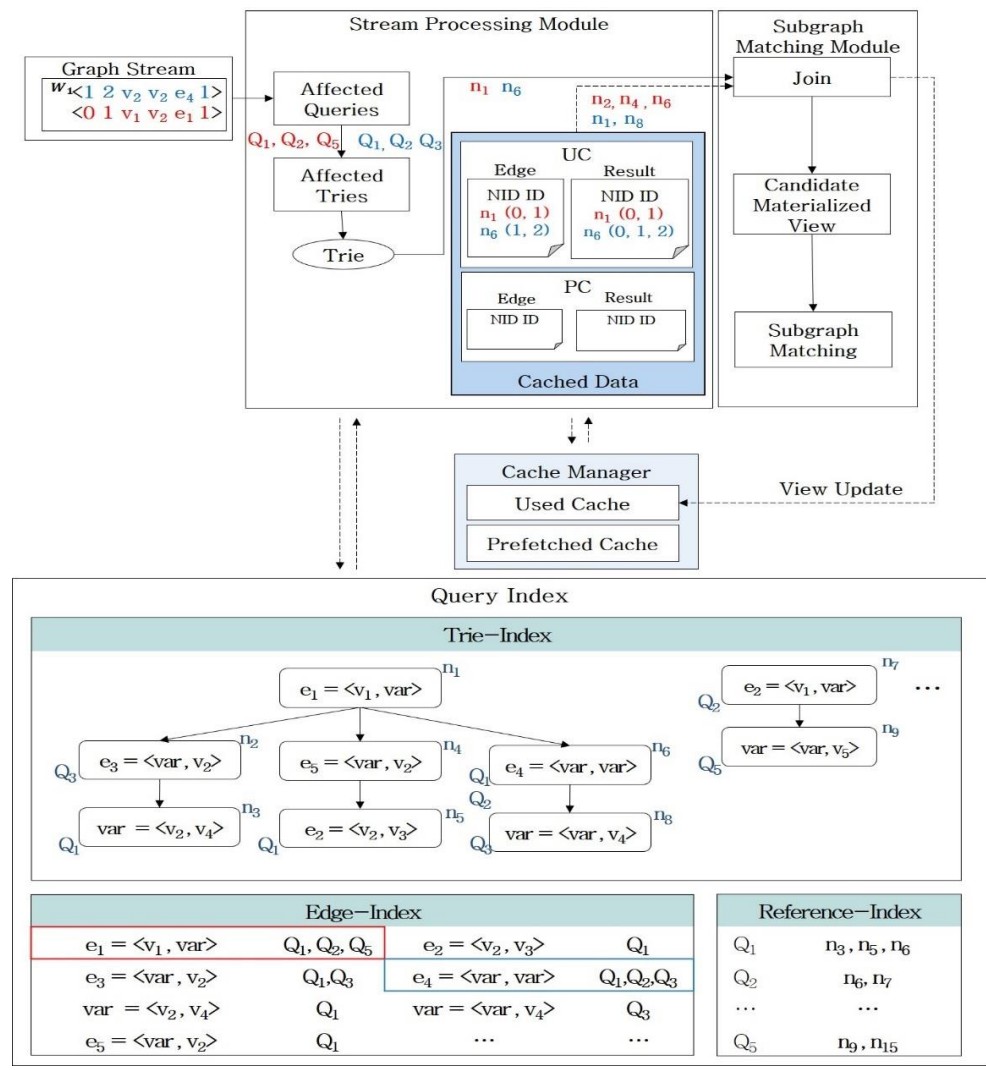

**Figure 8.** Procedure of subgraph matching.

Figure 9 shows the processes of continuous subgraph matching based on a graph stream under the application of a sliding window. After performing query processing, the Query Processing Manager presents the results derived by reflecting intermediate results in a cache and subgraph matching results. Then, it applies the FTV process based on trie indexing to the input stream data to maintain query processing results in the form of a materialized view according to trie nodes. It generates the final subgraph matching results based on the join operation results on trie nodes conducted in the reference indexing process. Specifically, it generates the final subgraph matching results {(0, 1, 2), (0, 3)} for $Q_2$ based on join operations on the result (0, 1, 2) of a materialized view based on the candidate for n6 and the result (0, 3) of a materialized view based on the candidate for n7.

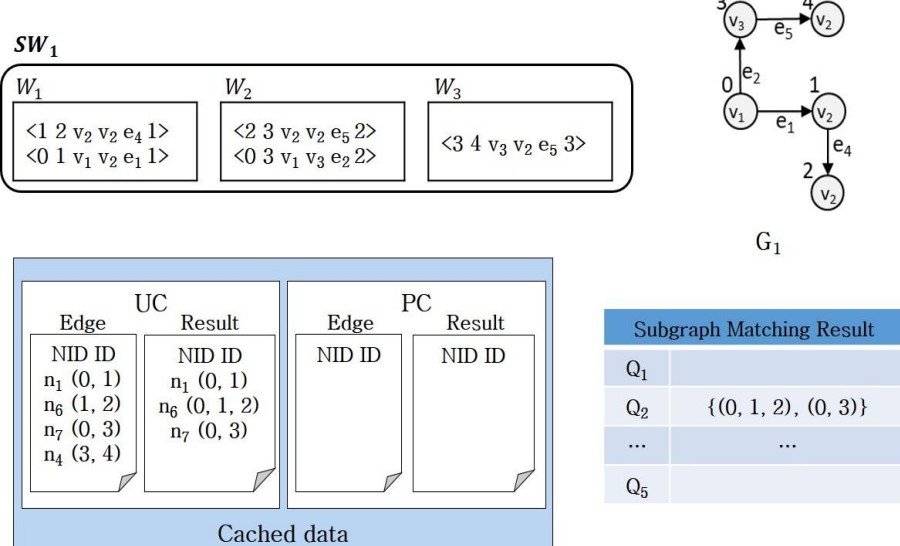

**Figure 9.** Results of a sliding window ($G_1$).

## 4. Performance Evaluation

In this study, the performance of the proposed continuous subgraph matching algorithm is compared with that of existing continuous subgraph matching algorithms. In terms of the environment for performance evaluation, an operating system with the specifications of Intel(R) Core(TM) i7-9700 CPU @ 3.6GHz 64bit was used. The maximum available memory was restricted to 1 GB. Python 3.6.8 was selected as the language to be used for the proposed scheme. Three data sets, wiki-Talk, cit-Patents, and YouTube, provided by SANP [25] were used in the performance evaluation. Table 2 shows the characteristics of the graphs used in the performance evaluation. Wiki-Talk indicates Wikipedia's communications network. Based on the contents of users' discussions, it displays information on users participating in discussions and articles upon which they developed discussions. Cit-Patents is a dataset of patents accumulated for 37 years in the United States. A graph was formed based on reference data in the cit-Patents dataset. As for the YouTube set, a community including the following three nodes was excluded. This set includes the graph data of communities including 13.5 or more nodes on average. This set includes information on approximately 8000 communities and shows a comparatively high clustering coefficient.

**Table 2.** Sets of experimental data.

| Dataset | # of Vertices | # of Edges |
|---|---|---|
| wiki-Talk | 2,394,385 | 5,021,410 |
| cit-Patents | 3,774,768 | 16,518,948 |
| YouTube | 1,134,890 | 2,987,624 |

Table 3 shows the parameters used in the performance evaluation. In this study, the performance of the proposed scheme was compared with that of an existing algorithm by adjusting the ratio of the used cache or prefetched catch to the entire cache space. To this end, the processing time was measured by changing the complexity of the query, the size of the window, and graph stream adjustment ratio. $\alpha$ refers to the ratio of the used cache to the entire cache. If $\alpha$ is 10, it indicates that the used cache accounts for 10% of the entire cache space and that the prefetched cache accounts for 90% of the entire cache space. To compare the data processing speed according to the complexity of the query, tests were conducted by varying a query as a simple query, complex vertex query, and complex vertex+edge query. Among the queries of interactive workload types provided by the LDBC (Linked Data Benchmark Council) [26], queries of a complex read-only query

type are used in this study. The queries of existing query types provided by the LDBC were defined as simple queries, and the queries including wildcard operations on vertices were defined as complex vertex queries. The queries including wildcard operations on vertices and edges were defined as complex vertex+edge queries. Moreover, tests were conducted by adjusting the size of the window from 100 to 1000 to measure the change in performance according to the size of the window. In each evaluation test, the size of the cache was adjusted from 0.1 to 1 kB to compare the performance according to the size of the cache. To measure the change in performance according to the graph stream adjustment ratio, tests were conducted by adjusting the graph stream adjustment ratio from 10% to 50%. Similarly, the size of the cache was adjusted from 0.1 to 1 kB in each evaluation test. Herein, the performance of the proposed scheme was compared with that of the existing scheme [20]. Since it is the subgraph matching scheme using the trie index, which is most similar to the proposed scheme among the existing schemes, it was selected as a comparison target. The existing scheme [20] efficiently performs continuous graph query processing based on a graph stream.

**Table 3.** Parameters for the performance evaluation.

| Name | Value |
| --- | --- |
| # of window size (# of vertices) | 100, 300, 500, 1000 |
| Cache Size(kB) | 0.1, 0.3, 0.5, 1 |
| $\alpha$ (%) | 10, 20, 30, 40, 50, 60, 70, 80, 90 |
| Graph Modification Ratio (%) | 10, 20, 30, 50 |

An independent performance evaluation was conducted to determine the optimal cache replacement strategy and optimal cache space ratio. Specifically, the window processing time required by the proposed cache replacement strategies was compared with that required by existing cache replacement strategies. The hit rate was measured according to the cache replacement strategies to identify the optimal cache replacement strategy. Moreover, the window processing time was compared according to the ratio of use of a two-level cache system. Figure 10 shows the average window processing time according to the cache replacement strategies when the size of the cache was established as 0.1 kB. A single window consisted of 100 edges and the average window processing time based on 500 windows was measured according to the datasets. With regard to cache replacement strategies, the proposed strategies (POP, POU, PUS, and PUSQ strategies) were compared with general strategies (LRU, first-in first-out (FIFO), and least frequently used (LFU) algorithms). When the proposed cache replacement strategies were applied to the wiki-Talk dataset, the average window processing time was 29.7 ms. This result confirmed that the proposed strategies increased the window processing performance by 182% compared with existing cache replacement strategies. When the proposed strategies were applied to the cit-Patents dataset, the window processing performance increased by 163%. When the proposed strategies were applied to the YouTube dataset, the window processing performance increased by 170%. Based on the measurement results, the proposed cache replacement strategies increased the window processing performance by 171% on average compared with existing algorithms. In terms of datasets, the PUSQ strategy derived the best performance for the processing of the wiki-Talk and YouTube datasets. The POP strategy derived the best performance for the processing of the cit-Patents dataset. This result indicates that the use of an appropriate cache replacement strategy for data characteristics can lead to efficient cache replacement.

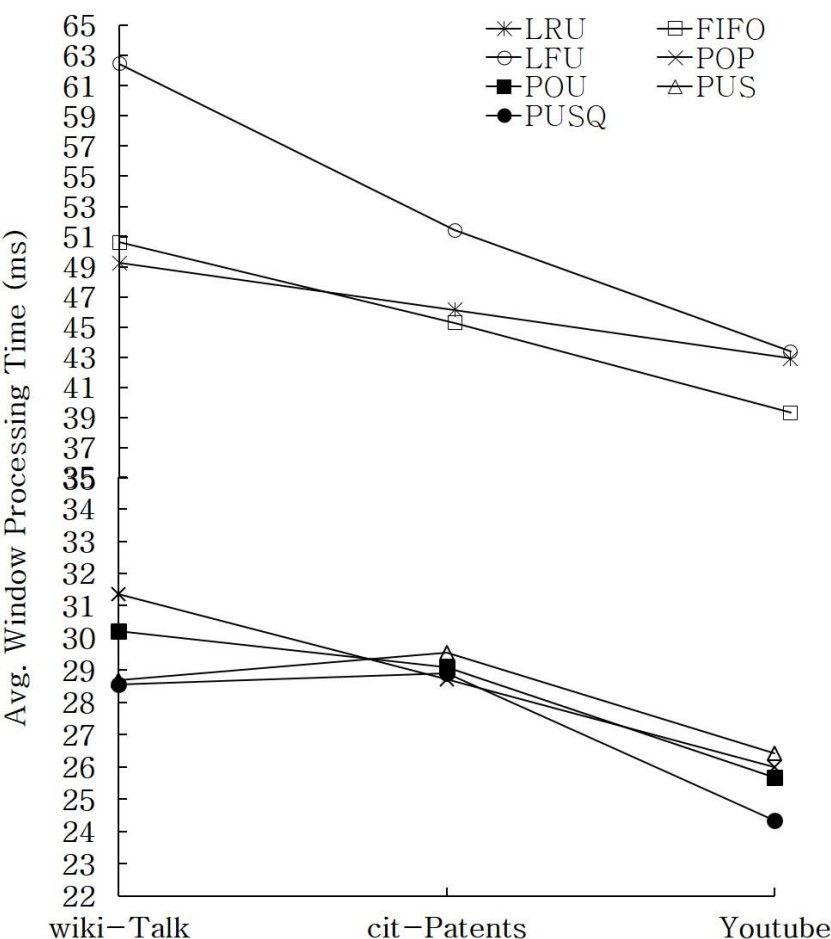

**Figure 10.** Average window processing time according to cache replacement strategies.

Figure 11 shows the hit rate according to the cache replacement strategies when the size of the cache was established as 0.1 kB. A single window includes 100 edges, and an analysis of the hit rate based on 100 windows was conducted. The analytical result based on the wiki-Talk dataset showed that the PUSQ strategy derived the highest hit rate of 70.29%. The hit rates for the used cache and prefetched cache were 43.25% and 27.04%, respectively. The analytical result based on the cit-Patents dataset showed that the POP strategy derived the highest hit rate of 66.28%. The hit rates for the used cache and prefetched cache were 27.99% and 38.30%, respectively. The analytical result based on the YouTube dataset showed that the PUSQ strategy derived the highest hit rate of 67.90%. The hit rates for the used cache and prefetched cache were 43.93% and 23.98%, respectively. As the PUSQ strategy derived the best performance on average, in this study, this algorithm is established as the main cache replacement strategy to be used to compare the performance of the proposed continuous subgraph matching scheme with that of the existing algorithm.

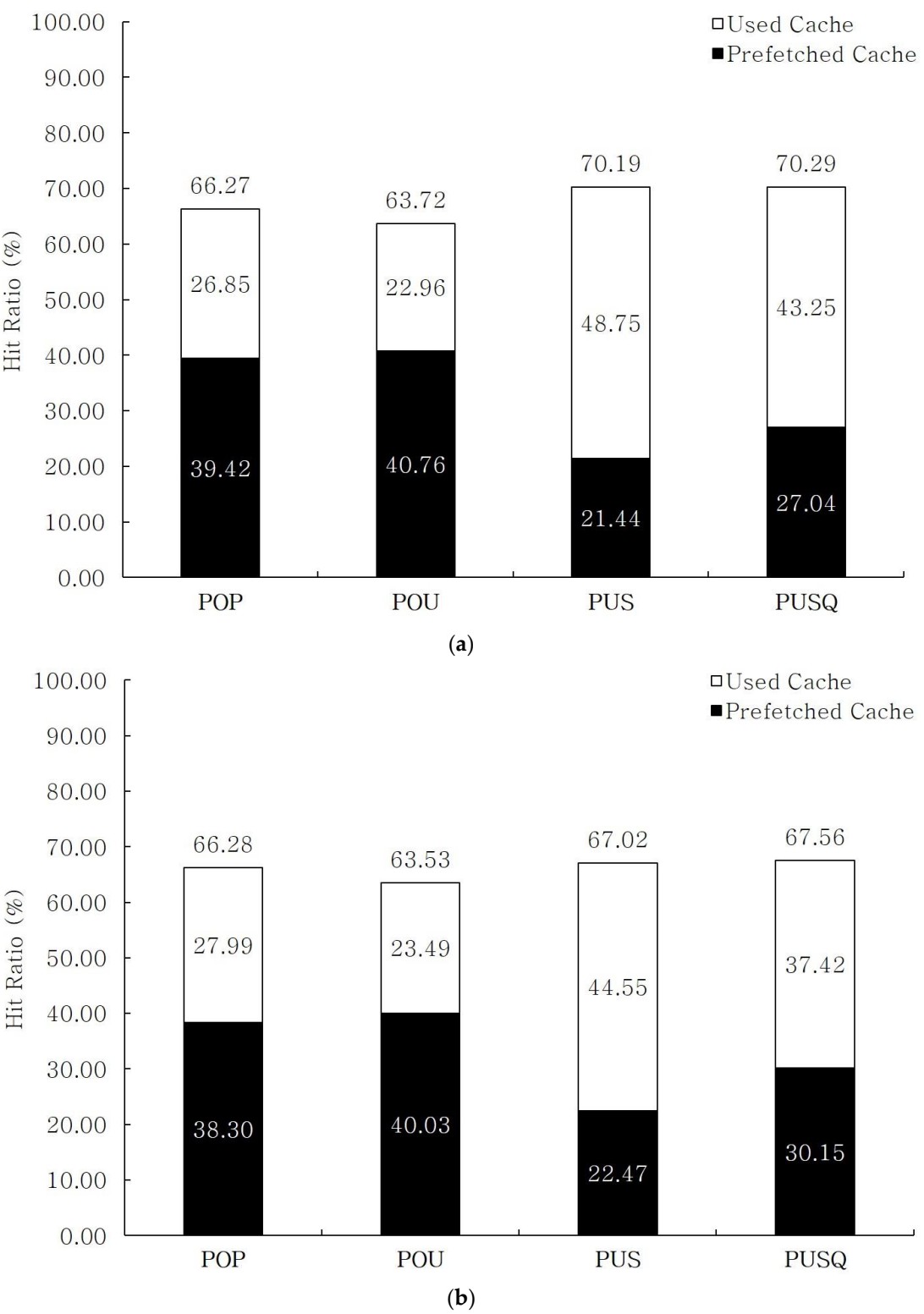

**Figure 11.** *Cont.*

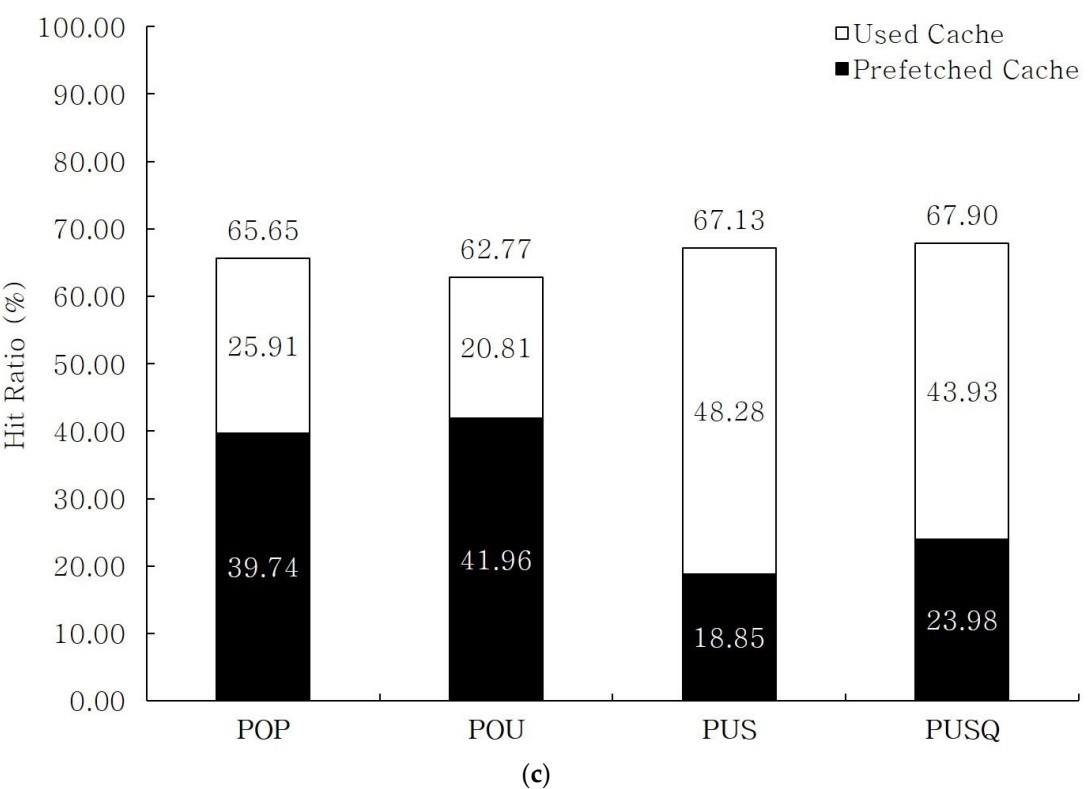

**Figure 11.** Hit rates according to the cache replacement strategies: (**a**) Wiki-Talk; (**b**) Cit-Patents; (**c**) YouTube.

Figure 12 shows the stream processing time as compared to the worst performance derived from the PUSQ cache replacement strategy according to different cache space ratios when the size of the cache was 0.1 kB. A single window contained 100 edges, and the processing time was measured based on 500 windows. The longest window processing time was required when $\alpha$ was 90%. Hence, the window processing time was established as 100% when $\alpha$ was 90%. The relevant window processing time is presented according to a change in the $\alpha$ value. The proposed algorithm derived the best window processing performance based on the wiki-Talk, cit-Patents, and YouTube datasets when the $\alpha$ values were 40%, 10%, and 30%, respectively. In addition, the performance evaluation result showed a difference in the performance of a two-level cache system according to the datasets. Based on this result, it was analyzed that the cache space ratio should be adjusted to achieve the optimal query processing performance according to the characteristics of the datasets.

In this study, the performance of the proposed continuous subgraph matching scheme was compared with that of an existing algorithm. To this end, the processing time required by both algorithms was compared by changing the complexity of the query, the size of the window, and graph stream adjustment ratio. Figure 13 shows the ratio of increase in the processing speed according to the complexity of the query. The processing time was measured by changing the queries to examine the change in performance according to the complexity of the query. The PUSQ strategy was adopted as a cache replacement strategy in this evaluation, and $\alpha$ was established to derive the optimal performance according to the datasets. The graph adjustment ratio was established as 10%, and a single window included 100 edges. The processing time was measured based on 500 windows. The size of the cache was adjusted from 0.1 to 1 kB. The result processing time required by the existing algorithm was set as 100%, and the increase in processing speed was indicated as a percentage according to the size of the cache. The analytical result showed that the proposed scheme increased the average processing performance by 194% based on simple queries, by 193% based on complex vertex queries, and by 192% based on complex vertex + edge queries according to each dataset. Regardless of the query types, the redundancy of

queries and caching techniques serve as the main factors affecting the increase in processing performance. The evaluation result showed that the size of the cache exerted a more significant effect on the query processing performance than the complexity of the query in a system environment with limited memory. The proposed scheme increased the processing speed by 189%, 194%, 195%, and 194% when the size of the cache was 0.1, 0.3, 0.5, and 1 kB, respectively. This result indicated that the processing speed did not increase in proportion to an increase in the size of the cache. In this regard, it is analyzed that more efficient query processing can be achieved through the selection of an appropriate cache capacity. As mentioned earlier, we presented an efficient way to process queries by caching only the necessary information without maintaining the overall index in the memory. The validity of the proposed two-level caching technique was proved through performance evaluation according to the cache size.

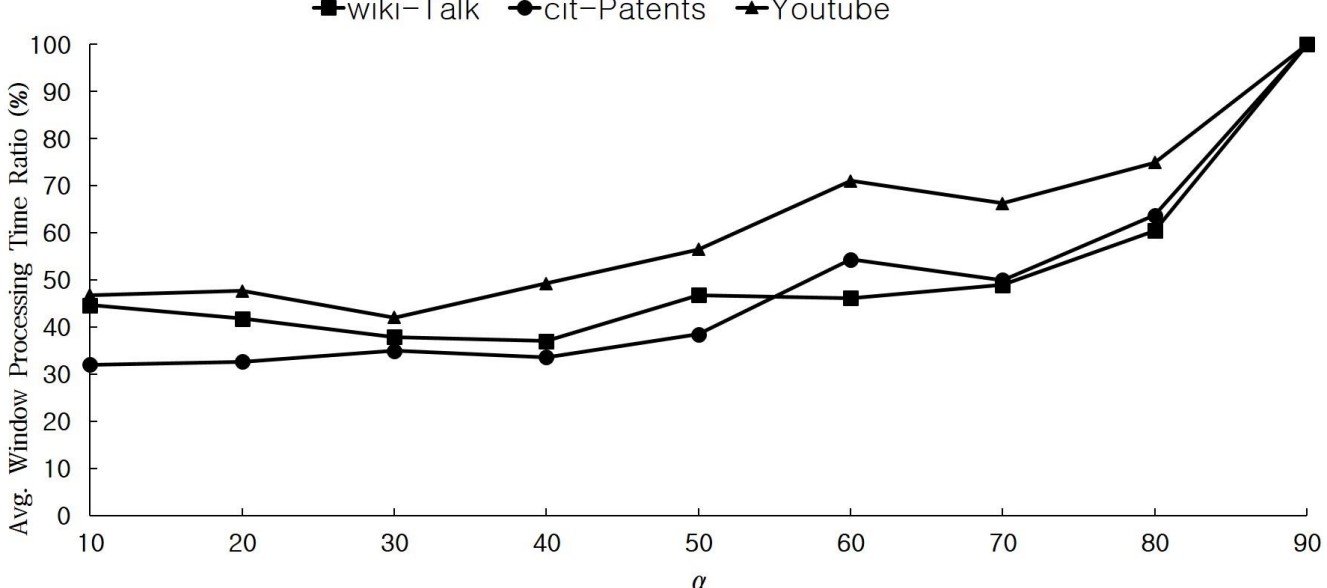

**Figure 12.** Average window processing time ratio according to $\alpha$ (the longest window processing time: $\alpha = 90$).

Figure 14 shows the average window processing time according to the size of the window. The window processing time was measured based on simple queries by establishing the graph adjustment ratio as 10% and adjusting the size of the window from 100 to 1000. Accordingly, the size of the cache was adjusted from 0.1 to 1 kB during the evaluation. The evaluation result showed that the proposed scheme increased the average window processing performance by 35% to 85% according to the size of the cache compared with the existing algorithm. The proposed scheme showed a linear increase in the average window processing time when the size of the window increased. The existing algorithm, which does not use an additional cache, showed a geometrical increase in the window processing time when the size of the window increased. In contrast, the proposed scheme, which applies a cache replacement strategy, showed a linear increase in the window processing time despite an increase in the size of the window.

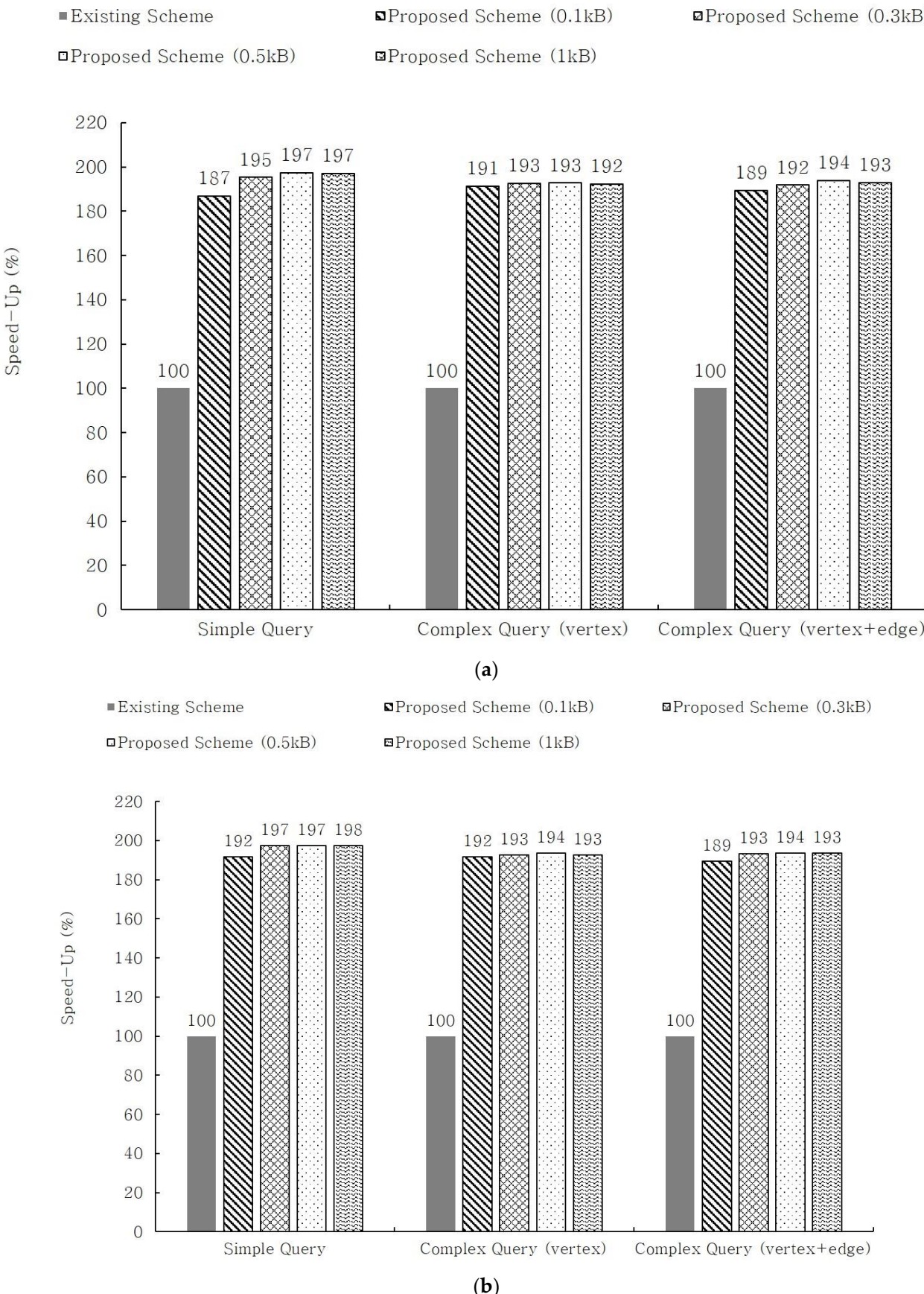

**Figure 13.** *Cont.*

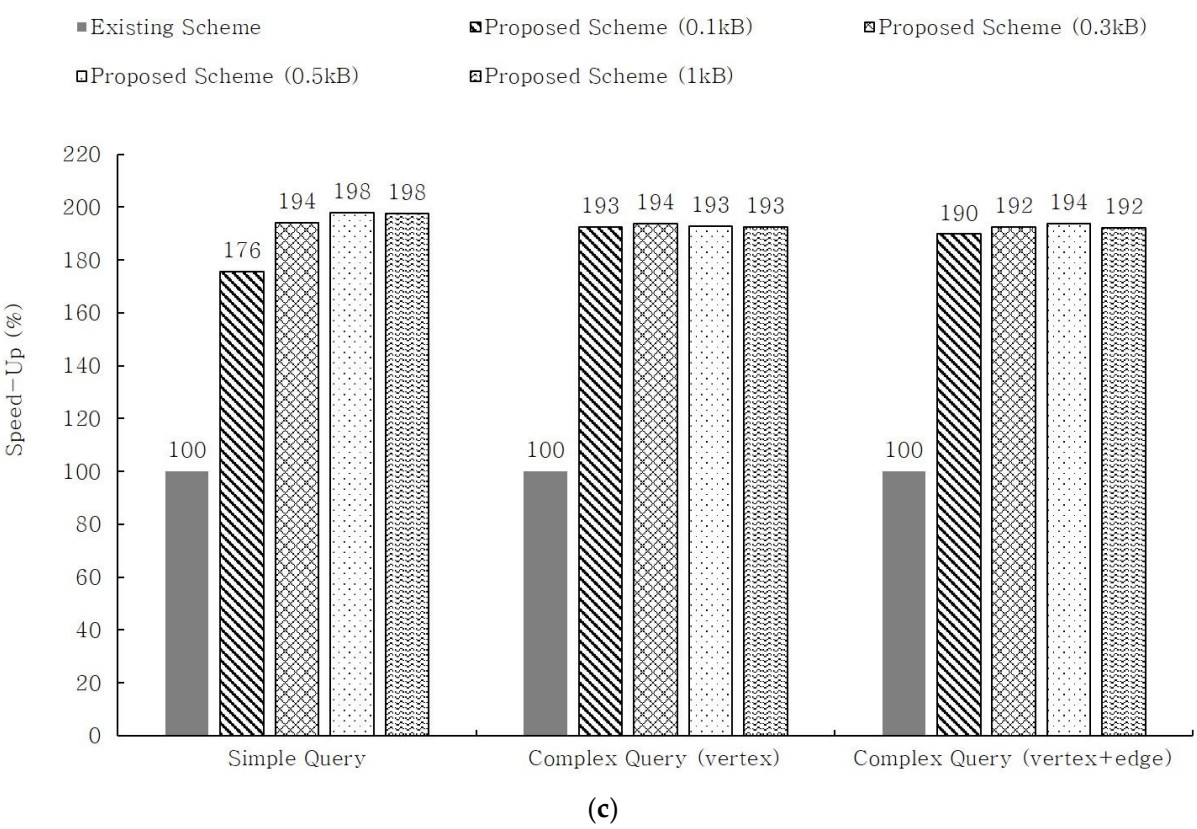

**(c)**

**Figure 13.** Ratios of increase in the processing speed according to the complexity of the query. (**a**) Wiki-Talk; (**b**) Cit-Patents; (**c**) YouTube.

Figure 15 shows the average window processing time according to the stream adjustment ratio. In this evaluation, the size of the window was considered as 100, and the window processing performance was compared based on 1000 windows. To measure the window processing performance according to the stream adjustment ratio accurately, the graphs of the initial 500 windows were input and the graph streams of the other 500 windows were adjusted. The window processing time was measured by varying the graph stream adjustment ratio as 10%, 20%, 30%, and 50%. The results of applying the existing and proposed algorithms to each dataset are as follows. The average window processing time of the existing algorithm increased by 206% when the stream adjustment ratio increased from 10% to 50%. The average window processing time of the proposed algorithm increased by 35% to 130% when the stream adjustment ratio increased from 10% to 50%. The existing algorithm requires a longer time to regenerate query results as the stream adjustment ratio increases. In contrast, the proposed algorithm exhibits a comparatively high processing speed owing to its function of managing a duplicated area. In addition, the window processing performance of the proposed algorithm based on the wiki-Talk dataset increased despite an increase in the graph stream adjustment ratio when the size of the cache was 0.5 kB. As indicated above the existing technique shows a delay of processing speed by twice when a stream adjustment ratio increased. In contrast, an increase in the graph stream adjustment ratio did not have a significant effect on the processing time except when the size of the cache was small.

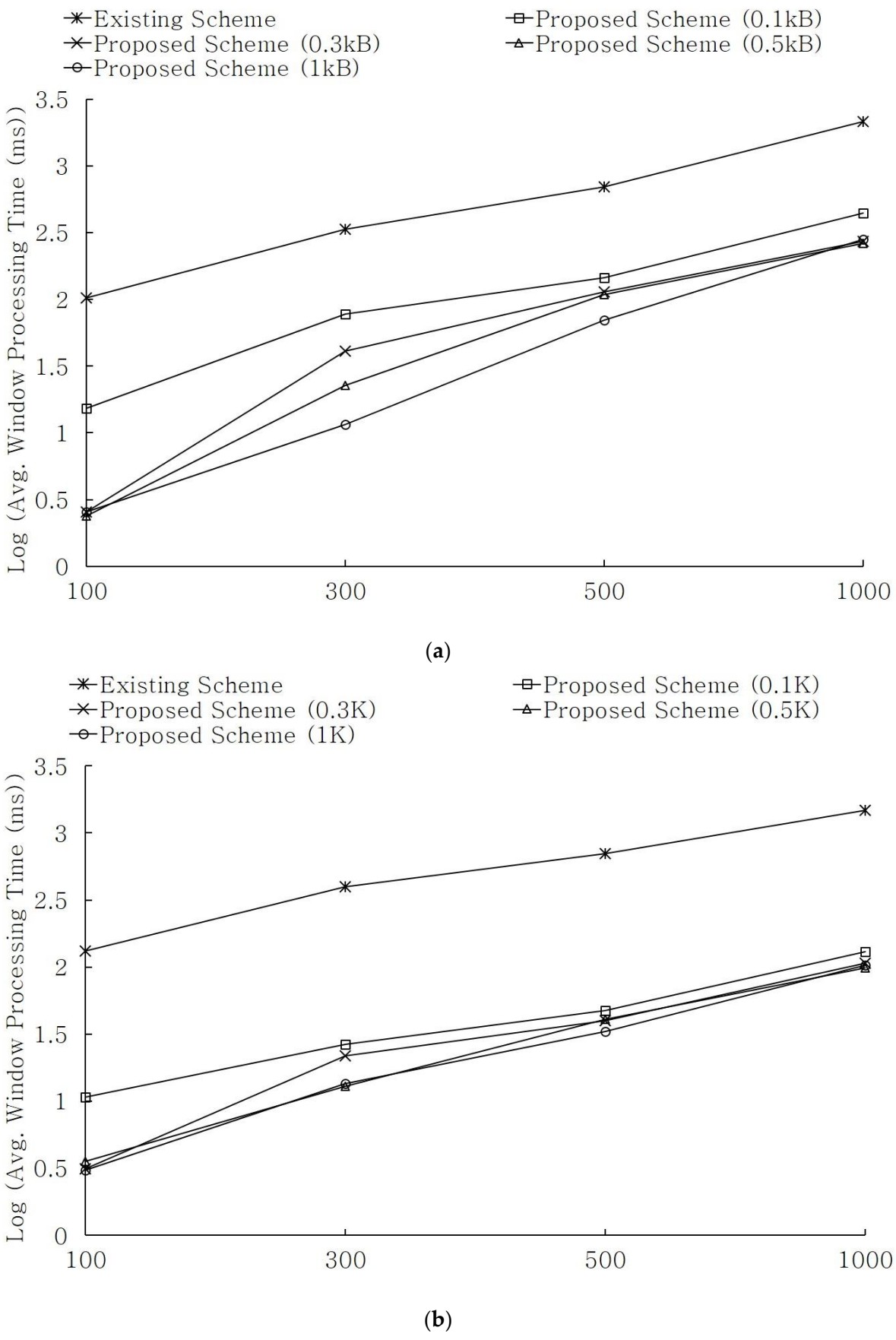

**Figure 14.** *Cont.*

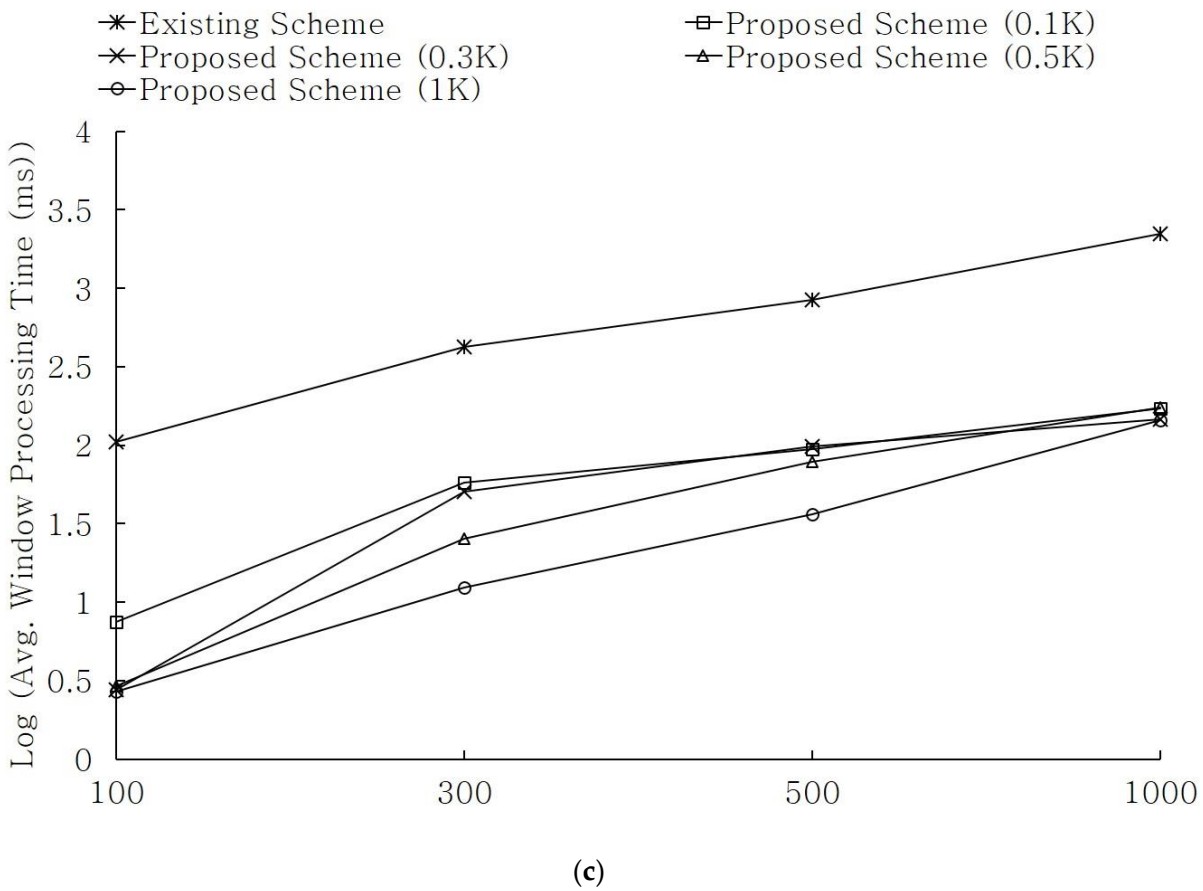

**Figure 14.** Average window processing time according to the size of the window. (**a**) Wiki-Talk; (**b**) Cit-Patents; (**c**) YouTube.

Figure 16 shows the memory cost according to the datasets. Performance comparisons between the existing scheme and the memory replacement techniques in the proposed scheme were performed. We measure the amount of memory (the number of bits) per time(ms) based on the simple query. As a result of the performance evaluation, the existing scheme used about four bits per ms, while the proposed cache replacement techniques used less than one bit on average. In terms of memory usage, it was confirmed that the PUSQ technique showed the best performance. In conclusion, the proposed cache replacement techniques have excellent performance over the existing scheme in terms of memory usage and query throughput.

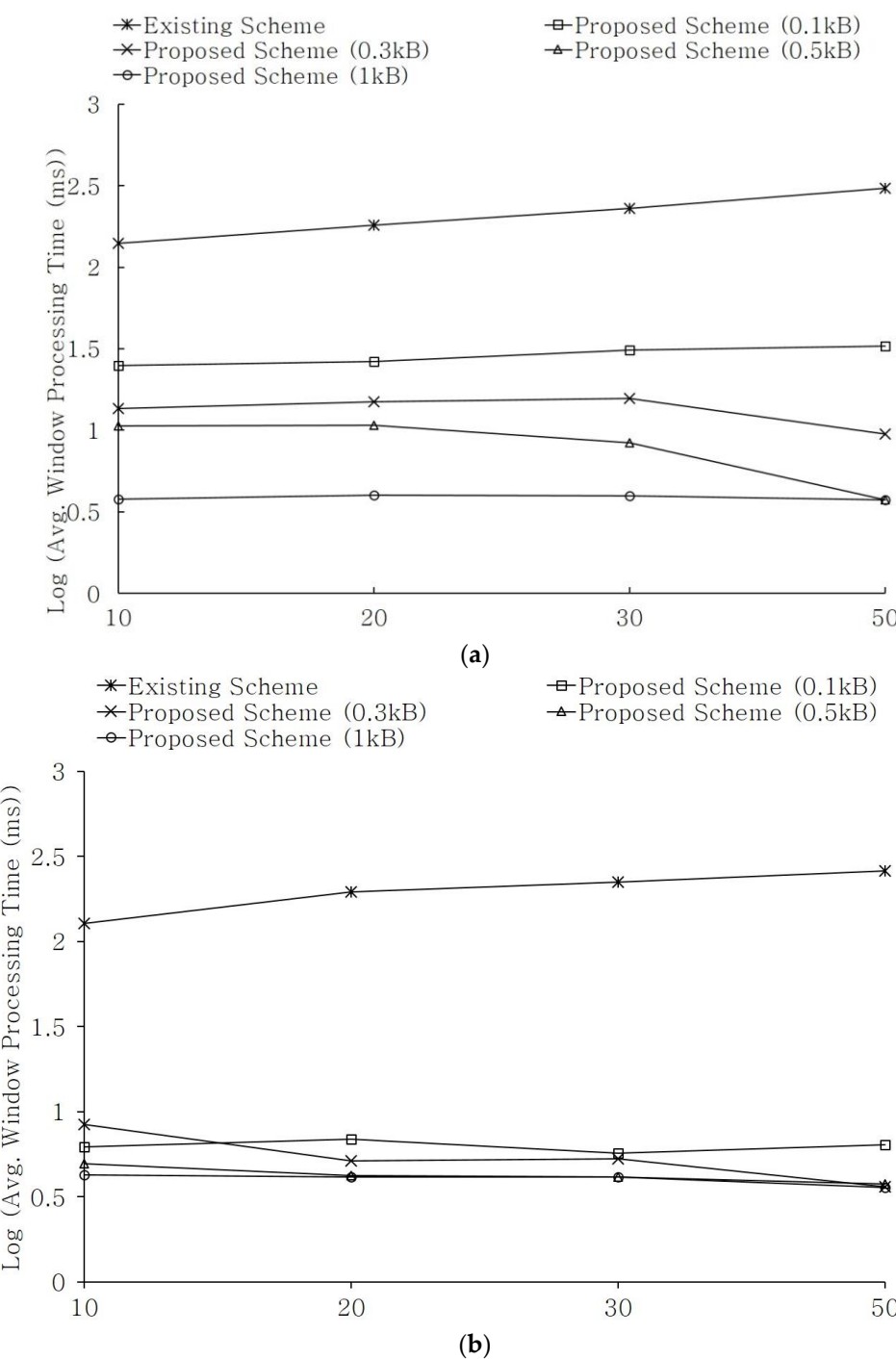

**Figure 15.** *Cont.*

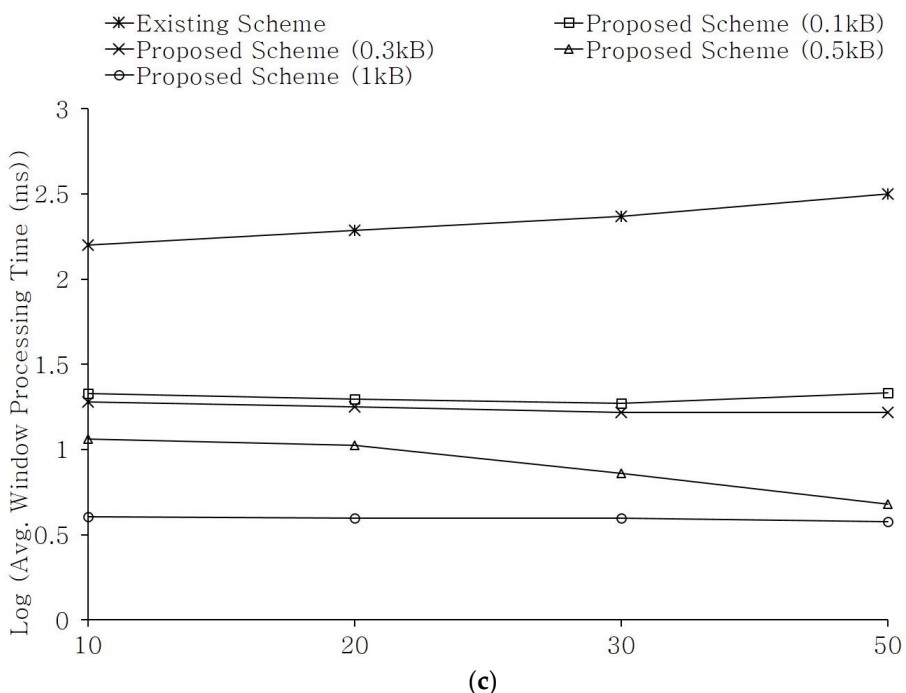

**Figure 15.** Average window processing time according to the stream adjustment ratio. (**a**) Wiki-Talk; (**b**) Cit-Patents; (**c**) YouTube.

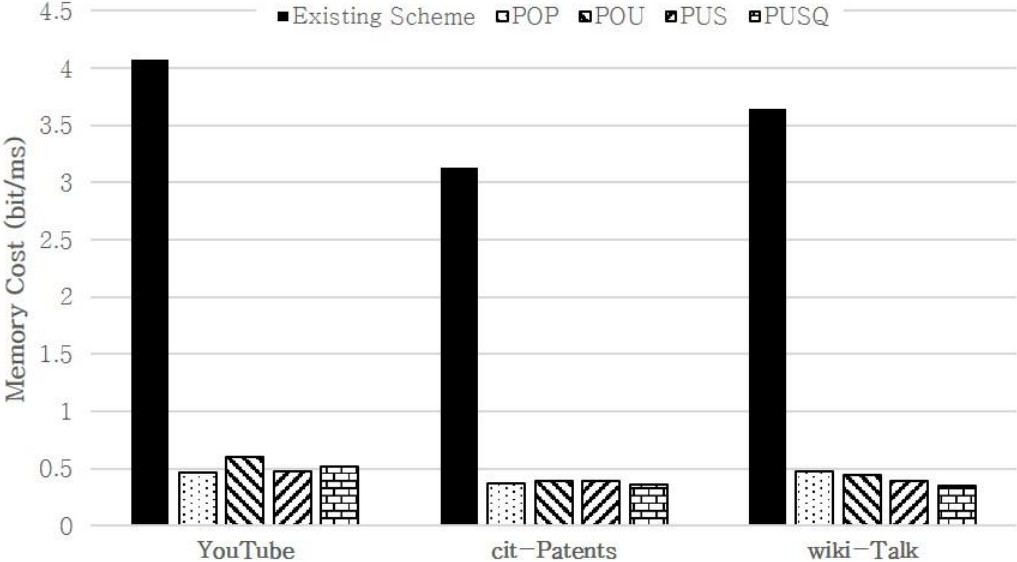

**Figure 16.** Memory cost according to datasets.

## 5. Conclusions

In this paper, a sliding-window-based continuous subgraph matching scheme was proposed to manage graph streams efficiently in a limited memory environment. The proposed scheme can respond to various types of queries by providing wildcard operations on vertices and edges. It identifies a duplicated query area and establishes an index for efficient continuous query processing. It also applies one-time indexing in the duplicated query area to generate several duplicated query results. Furthermore, it can efficiently manage intermediate query results by adopting a two-level catching algorithm. Moreover, cache replacement strategies based on statistics and indexing characteristics were developed, and efficient cache replacement was performed based on the developed algorithms. An independent performance evaluation revealed that efficient query processing can be achieved by the implementation of an appropriate cache replacement strategy and an

appropriate size of cache space according to the datasets. The evaluation result showed that the performance of the proposed scheme increased by 190% at the maximum compared with that of the existing algorithm. The proposed scheme significantly increased the query processing speed regardless of the complexity of the query and the size of the window. The proposed scheme can be used for various applications that analyze graph types frequently searched in SNS graph data, register network attack types as query graphs to detect network traffic types in real-time, and so on. Further research will be conducted to propose continuous query processing methods according to the input of query streams and processing methods that support various graph attributes. Compression techniques are very necessary in graph processing. Since various studies related to graph compression have continuously been conducting, we are sure that we can get the high performance of the proposed scheme by applying them to it in the future. Follow-up research will also verify the excellent performance of the proposed scheme based on additional comparisons with the existing algorithm.

**Author Contributions:** Conceptualization, D.C., S.L., H.L., J.L., K.B. and J.Y.; methodology, D.C., S.L., H.L., J.L., K.B. and J.Y.; validation, D.C., S.L., S.K., H.L., J.L. and K.B.; formal analysis, D.C., S.L., S.K., H.L., J.L. and K.B.; writing—original draft preparation, D.C., S.L., S.K. and K.B.; writing—review and editing, J.Y. All authors have read and agreed to the published version of the manuscript.

**Funding:** This research was funded by the National Research Foundation of Korea (NRF) grant funded by the Korea government (MSIT). (No. 2022R1A2B5B02002456), by Cooperative Research Program for Agriculture Science and Technology Development (Project No. PJ016247012023) Rural Development Administration, The Research Year of Chungbuk National University in 2019, and by the MSIT (Ministry of Science and ICT), Korea, under the Grand Information Technology Research Center support program (IITP-2023-2020-0-01462) supervised by the IITP (Institute for Information & communications Technology Planning & Evaluation).

**Institutional Review Board Statement:** Not applicable.

**Informed Consent Statement:** Not applicable.

**Data Availability Statement:** Not applicable.

**Conflicts of Interest:** The authors declare no conflict of interest.

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
