# Peer review of "Efficient Continuous Subgraph Matching Scheme Based on Trie Indexing for Graph Stream Processing"

_applsci, doi:10.3390/app13085137_

Round 1

Reviewer 1 Report

This paper proposes a new method to detect subgraphs from graph streams. This problem is applicable to a variety of applications where graphs are used, such as social networks and transportation networks. The proposed method improves upon the existing methods in terms of query processing time and memory usage. The results show the query processing times are low.

This article is lacking in two ways: literature review and evaluations.

It is unclear based on the literature review how the article aims to address the gaps in previous work. For instance, Section 2 describes many aspects of related methods, but never appropriately points out their limitations and how the authors will address them in this paper. The most related article seems to be Reference 20. But, it is not very clear how it differs from current work. The authors should rewrite Section 2 with the goal of organizing the related literature in a way that explicitly shows how the new methods address the limitations. Otherwise, it is not easy to evaluate the new contributions. Moreover, I recommend that authors include an itemized list of contributions in the Introduction section.

Memory cost evaluations are missing from the article. All evaluations are on processing times. The authors claim that they have made improvements in memory requirements. This claim needs to be validated in some way. Either by comparing the current methods to the literature, or by evaluating the current method with and without the design decisions.

There is no indication that the proposed method is compared with any methods from the literature. If that is the case, the results are not acceptable. However, if the authors have compared with the literature, they must cite them. Just saying "existing scheme" is not acceptable (Figures 13-15). The authors need to dedicate some text to introduce and cite a collection of existing methods (just one is not enough) and say why they are the most appropriate for comparison. I was unable to find a citation for the "existing algorithm" the authors mention repeatedly.

Moreover, in the current results, some choices of plots are inappropriate. For instance, in Figure 10, the horizontal axis has no order. But the result is presented with a line plot. Meanwhile, in Figure 12, the horizontal axis is a real number and the authors have chosen to use a box plot.

Overall, the novelty and contributions are unclear, and the evaluations are incomplete and lack comparison with literature.

Author Response

We would like to sincerely thank you for your attentive indications and good comments. Our paper is partially rewritten in order to revise and complement your comments. Please refer to the attached file, named "Response to Reviewer1's comments(somin).docx

Many thanks.

Jaesoo Yoo

Reviewer 2 Report

The paper provides a good overview of related work and the current status of problem solving. Reference fully reflects the content of the work.

The processing of data graphs is certainly an urgent problem due to the spread of the use of network structures.

There are no significant comments on the work. If possible, please add your thoughts on the use of compression algorithms in the process of processing graph streams. Is it necessary and advisable? What are the prospects in this direction?

Author Response

We would like to sincerely thank you for your attentive indications and good comments. Our paper is partially rewritten in order to revise and complement your comments. Please refer to the attached file, named "Response to Reviewer2's comments(somin).docx

Many thanks.

Jaesoo Yoo

Reviewer 3 Report

Dear colleagues,

Your article is really interesting for researchers, working in the domain graph processing algorithms but you can expand the circle of readers if you add a use case that will show the possibility of practical application of the proposed approach and its benefits for solving real life problems.

Author Response

We would like to sincerely thank you for your attentive indications and good comments. Our paper is partially rewritten in order to revise and complement your comments. Please refer to the attached file, named "Response to Reviewer3's comments(somin).docx

Many thanks.

Jaesoo Yoo

Reviewer 4 Report

Congratulations on this interesting work

In order to improve the quality of your paper, please take into account the following remarks: 

Authors should show their main contribution to the work.

The methodology, the proposed module and the results are very interesting and rich 

Particular attention should be paid to the quality of the graphics used

Good luck 

Author Response

We would like to sincerely thank you for your attentive indications and good comments. Our paper is partially rewritten in order to revise and complement your comments. Please refer to the attached file, named "Response to Reviewer4's comments(somin).docx

Many thanks.

Jaesoo Yoo

Round 2

Reviewer 1 Report

The authors have made significant improvements to the paper. However, I still think this paper needs memory cost evaluation. Without such evaluations the value of the contributions is significantly diminished. Therefore, I recommend a major revision, with enough time to allow the new evaluations to be done.

Author Response

Dear reviewer,

Thank your for your precious second comments. We did our best to reflect your comments about memory cost. Please refer to the attached file. We are really hoping you can accept our paper for publication in Applied Sciences. 

Many thanks.

Jaesoo Yoo
